

# Virus discovery in all three major lineages of terrestrial arthropods highlights the diversity of single-stranded DNA viruses associated with invertebrates

Karyna Rosario[1], Kaitlin A. Mettel[1], Bayleigh E. Benner[1],
Ryan Johnson[1], Catherine Scott[2], Sohath Z. Yusseff-Vanegas[3],
Christopher C.M. Baker[4,5], Deby L. Cassill[6], Caroline Storer[7],
Arvind Varsani[8,9] and Mya Breitbart[1]

[1] College of Marine Science, University of South Florida, Saint Petersburg, FL, USA
[2] Department of Biological Sciences, University of Toronto, Scarborough, Scarborough, ON, Canada
[3] Department of Biology, University of Vermont, Burlington, VT, USA
[4] Department of Ecology and Evolutionary Biology, Princeton University, Princeton, NJ, USA
[5] Department of Organismic and Evolutionary Biology, Harvard University, Cambridge, MA, USA
[6] Department of Biological Sciences, University of South Florida Saint Petersburg, Saint Petersburg, FL, USA
[7] School of Forest Resources and Conservation, University of Florida, Gainesville, FL, USA
[8] The Biodesign Center for Fundamental and Applied Microbiomics, School of Life Sciences, Center for Evolution and Medicine, Arizona State University, Tempe, AZ, USA
[9] Structural Biology Research Unit, Department of Clinical Laboratory Sciences, University of Cape Town, Cape Town, South Africa

Corresponding author
Karyna Rosario,
krosari2@mail.usf.edu

## ABSTRACT

Viruses encoding a replication-associated protein (Rep) within a covalently closed, single-stranded (ss)DNA genome are among the smallest viruses known to infect eukaryotic organisms, including economically valuable agricultural crops and livestock. Although circular Rep-encoding ssDNA (CRESS DNA) viruses are a widespread group for which our knowledge is rapidly expanding, biased sampling toward vertebrates and land plants has limited our understanding of their diversity and evolution. Here, we screened terrestrial arthropods for CRESS DNA viruses and report the identification of 44 viral genomes and replicons associated with specimens representing all three major terrestrial arthropod lineages, namely Euchelicerata (spiders), Hexapoda (insects), and Myriapoda (millipedes). We identified virus genomes belonging to three established CRESS DNA viral families (*Circoviridae*, *Genomoviridae*, and *Smacoviridae*); however, over half of the arthropod-associated viral genomes are only distantly related to currently classified CRESS DNA viral sequences. Although members of viral and satellite families known to infect plants (*Geminiviridae*, *Nanoviridae*, *Alphasatellitidae*) were not identified in this study, these plant-infecting CRESS DNA viruses and replicons are transmitted by hemipterans. Therefore, members from six out of the seven established CRESS DNA viral families circulate among arthropods. Furthermore, a phylogenetic analysis of Reps, including endogenous viral sequences, reported to date from a wide array of organisms revealed that most of the known CRESS DNA viral diversity circulates among invertebrates. Our results highlight the vast and

unexplored diversity of CRESS DNA viruses among invertebrates and parallel findings from RNA viral discovery efforts in undersampled taxa.

## INTRODUCTION

Virus discovery remains an open-ended endeavor with estimates of more than 99% of viruses within organisms remaining to be sampled (*Geoghegan & Holmes, 2017*). Eukaryotic viruses infecting vertebrates, mainly mammals, and land plants are overrepresented in public databases relative to those infecting invertebrates and unicellular organisms (*Mahmoudabadi & Phillips, 2018*). Therefore, biased sampling has heavily skewed our view of viral diversity and evolution and there is a need to explore "non-traditional" organisms. Efforts investigating single-stranded (ss)RNA viruses in undersampled taxa have identified arthropods, the most diverse and successful group of animals on Earth (*Stork, 2018*), as a rich and untapped reservoir of novel viruses (*Li et al., 2015*; *Shi et al., 2016a*). Moreover, discovery of divergent viruses in invertebrates has prompted reevaluation of RNA virus evolution concepts and taxonomic frameworks (*Dolja & Koonin, 2018*; *Shi et al., 2016b*, *Shi, Zhang & Holmes, 2018b*). Notably, these studies have identified arthropods as the ultimate ancestral source of some vertebrate- and plant-infecting RNA viruses (*Shi et al., 2016a*). Since arthropods may be central to the evolutionary history of other viral groups, here we survey terrestrial arthropods for the presence of ssDNA viruses with circular genomes, which follow positive-sense ssRNA viruses as the second most abundant group of viruses infecting eukaryotes (*Mahmoudabadi & Phillips, 2018*; *NCBI, 2018*). This study focuses on a subset of eukaryotic ssDNA viruses with covalently-closed circular genomes that encode a replication-associated protein (Rep).

Prior to 2009, eukaryotic circular Rep-encoding ssDNA (CRESS DNA) viruses were thought to be restricted to plants (*Geminiviridae* and *Nanoviridae* families) and vertebrates (family *Circoviridae*), specifically pigs and birds (*Lefkowitz et al., 2018*). Since then, metagenomic studies have revealed the cosmopolitan and diverse nature of eukaryotic CRESS DNA viruses. CRESS DNA viruses have now been reported from a wide array of organisms, ranging from primates (*Kapusinszky et al., 2017*; *Ng et al., 2015*) to unicellular algae (*Yoon et al., 2011*), and ecosystems spanning aquatic (*Dayaram et al., 2015a*; *Hewson et al., 2013a*; *Labonté & Suttle, 2013*; *Lopez-Bueno et al., 2009*), terrestrial (*Kim et al., 2008*; *Reavy et al., 2015*), airborne (*Whon et al., 2012*) and man-made environments (*Kraberger et al., 2015*; *Rosario, Duffy & Breitbart, 2009*; *Rosario et al., 2018*). The increased detection and expanded diversity of CRESS DNA viruses has resulted in the establishment of four new taxonomic groups by the International Committee for the Taxonomy of Viruses, including three new families (*Genomoviridae*, *Smacoviridae*, *Bacilladnaviridae*) and the *Cyclovirus* genus within the family *Circoviridae*, to accommodate these diverse

viruses (*Kazlauskas et al., 2017*; *Krupovic et al., 2016*; *Rosario et al., 2017*; *Varsani & Krupovic, 2018*). Moreover, many CRESS DNA viruses, which are predicted to represent novel families, remain unclassified. In addition, the investigation of endogenous viral sequences has revealed the ancient origin of CRESS DNA viruses by providing evidence indicating that some of these viruses have been infecting diverse animal and plant hosts for millions of years (*Belyi, Levine & Skalka, 2010*; *Dennis et al., in press*; *Lefeuvre et al., 2011*). Although integration into host chromosomes may be incidental (*Krupovic & Forterre, 2015*), CRESS DNA endogenous viral elements may influence host evolution and biology by contributing to their genetic composition and, perhaps, providing new functional capabilities (*Belyi, Levine & Skalka, 2010*; *Feschotte & Gilbert, 2012*). Therefore, eukaryotic CRESS DNA viruses are a highly diverse group of viruses that have implications well beyond their recognized agricultural and medical relevance.

All eukaryotic CRESS DNA viruses are minimalists; their small circular genomes (<6 kb) encode <8 proteins, including a distinctive homologous Rep (*Kazlauskas et al., 2017*; *Rosario, Duffy & Breitbart, 2012*). Another salient feature of most CRESS DNA viral genomes is a conserved putative origin of replication (*ori*) marked by a nonanucleotide motif at the apex of a predicted stem-loop structure where rolling circle replication (RCR) is initiated (*Rosario, Duffy & Breitbart, 2012*). The presence of a capsid-encoding ORF (open reading frame (ORF)) distinguishes CRESS DNA viruses from CRESS DNA satellite molecules or replicons, such as those classified within the family *Alphasatellitidae* (*Briddon et al., 2018*). Although these circular molecules do not encode a capsid, a hallmark defining feature of viruses, these replicons have been considered part of the "extended viral world" as these molecules represent successful genetic parasites (*Koonin & Dolja, 2014*).

It has been hypothesized that CRESS DNA viruses may have evolved from interactions between capsid protein genes from RNA viruses and bacterial plasmids on several independent occasions (*Koonin, Dolja & Krupovic, 2015*; *Krupovic, 2013*). The polyphyletic nature of CRESS DNA viruses, complemented by their high substitution rates (*Duffy & Holmes, 2009*; *Duffy, Shackelton & Holmes, 2008*; *Firth et al., 2009*) and predisposition to recombination (*Lefeuvre et al., 2009*; *Martin et al., 2011*), even within the *rep* gene (*Kazlauskas, Varsani & Krupovic, 2018*; *Krupovic et al., 2015*), have resulted in the emergence of a highly diverse viral group. This diversity is also reflected by different genome architectures that, similar to RNA viruses (*Li et al., 2015*; *Shi et al., 2016a*), suggest plasticity in CRESS DNA virus genomes. CRESS DNA viruses, including viruses classified within the same genus (e.g., *Begomovirus*), can have monopartite or multipartite genomes. Notably, multipartite genomes have only been observed in plant-infecting CRESS DNA viruses. Based on the arrangement of major ORFs relative to the putative *ori*, CRESS DNA genomes display eight genome organizations, including those that only encode a Rep and might represent segments of multipartite genomes or satellite molecules (*Rosario, Duffy & Breitbart, 2012*). However, there does not seem to be a correlation between these genome organizations and phylogenetic relationships amongst various CRESS DNA viruses (*Quaiser et al., 2016*; *Rosario et al., 2015a*). The evolutionary history of some CRESS DNA viruses is further

obscured by rampant gene fragment exchanges that have led to chimeric Rep sequences that hinder taxonomic classification (*Kazlauskas, Varsani & Krupovic, 2018*). Despite these limitations, the Rep remains the only tractable phylogenetic marker that can be used to investigate evolutionary relationships among the highly diverse and polyphyletic CRESS DNA viruses.

Most viral discovery studies focus on vertebrate hosts, primarily mammals, which limits our perspective of viral diversity and evolution. The work presented here expands on studies investigating CRESS DNA viruses in invertebrates, which have traditionally been undersampled (*Bettarel et al., 2018*; *Bistolas et al., 2017*; *Dayaram et al., 2013*; *Hewson et al., 2013b*; *Kraberger et al., 2018*; *Rosario et al., 2012*, *2015a*; *Wang et al., 2018*). We report 44 CRESS DNA genomes recovered from arthropods representing all three major terrestrial arthropod lineages (*Giribet & Edgecombe, 2012*). By performing a phylogenetic analysis of Reps from CRESS DNA genomes reported from a wide array of organisms and those identified as endogenous viral elements, we demonstrate that most of the previously described CRESS DNA viral phylogenetic diversity circulates among invertebrates. In addition, database searches using newly detected Reps led to the detection of an unreported endogenous cyclovirus-like element within a genome scaffold from a rodent-infecting nematode. Although cycloviruses have been mainly detected in feces from various mammals and homogenized tissues from insects (*Rosario et al., 2017*), endogenous cyclovirus elements indicate that these viruses are able to infect both arthropod (mites) (*Dennis et al., 2018*; *Liu et al., 2011*) and non-arthropod parasitic invertebrates.

## MATERIALS AND METHODS

### Sample collection and processing

A variety of opportunistically sampled arthropods were screened for CRESS DNA viral sequences (Table 1). Samples included members from all three major groups of terrestrial arthropods including Hexapoda (Class Insecta; Orders: Hymenoptera, Coleoptera, Odonata, Dermaptera, Diptera, Orthoptera, Lepidoptera, Ephemeroptera, Blattodea), Euchelicerata (Class Arachnida; Order: Araneae), and Myriapoda (Classes: Diplopoda and Chilopoda). All specimens were identified to the most specific taxonomic rank possible through identification by experts or using DNA barcoding (see below) when taxonomic identifications were not available. Samples were processed following methods used to detect CRESS DNA viruses in marine invertebrates (*Rosario et al., 2015a*) and insects (*Dayaram et al., 2013*; *Rosario et al., 2012*). Briefly, specimens were serially rinsed three times using sterile suspension medium (SM) buffer [0.1M NaCl, 50 mM Tris–HCl (pH 7.5), 10 mM MgSO$_4$] to remove debris. A small piece of tissue was dissected from representative specimens and stored at −20 °C for DNA barcoding. Each specimen or pooled sample composed of up to 10 specimens from the same species was homogenized in SM buffer through bead-beating using 1.0 mm sterile glass beads in a bead beater (Biospec Products, Bartlesville, OK, USA) for 60–90 s and homogenates were centrifuged at 6,000 × $g$ for 6 min. Viral particles were then partially purified from supernatants by filtering through a 0.45 μm Sterivex filter

**Table 1 Sample information and identified CRESS DNA genomes.**

| Year | Location[1] | Species name (common name)[2] | Samples[3] | Identified genomes |
|---|---|---|---|---|
| 2011 | Kenya | *Crematogaster nigriceps* (Arboreal ant) | Pool (2) | Arboreal ant associated circular virus 1 |
| 2011 | Kenya | *Tetraponera penzigi* (Arboreal ant) | Pool (2) | Arboreal ant associated circular virus 1 |
| 2011 | Kenya | *Crematogaster mimosae* (Arboreal ant) | Pool | Arboreal ant associated circular virus 1 |
| 2013 | FL USA | *Solenopsis invicta* (Fire ant) | Pool | Fire ant associated circular virus 1 |
| 2013 | FL USA | *Xylosandrus amputates* (Bark beetle) | Pool | Bark beetle associated circular virus 1 |
| 2014 | Puerto Rico | *Dineuteus* sp. (Water beetle)* | Single (4) | Water beetle associated circular virus 1 |
| 2013 | Store | *Gryllus assimilis* (Field cricket) | Pool | Cricket associated circular virus 1 |
| 2011 | FL USA | *Romulea microptera* (Lubber grasshopper) | Single | Grasshopper associated circular virus 1 |
| 2013 | Nevis | *Lucilia rica* (Blow fly)* | Pool | Fly associated circular virus 1 |
| | | | | Fly associated circular virus 3 |
| | | | | Fly associated circular virus 5 |
| 2013 | St. Barts | *Fannia* sp. (Dung fly)* | Pool | Fly associated circular virus 2 |
| 2013 | Dom. Republic | *Lucilia retroversa* (Blow fly) | Pool | Fly associated circular virus 4 |
| 2013 | Guadeloupe | *Lucilia rica* (Blow fly)* | Pool | Fly associated circular virus 6 |
| 2013 | St. Barts | *Lucilia rica.* (Blow fly)* | Pool | Fly associated circular virus 7 |
| 2015 | NH USA | *Oxidus* sp. (Greenhouse millipede)* | Single | Millipede associated circular virus 1 |
| 2017 | Victoria BC | *Parasteatoda tepidariorum* (Common house spider) | Single | Common house spider circular molecule 1 |
| 2017 | Victoria BC | *Cybaeus signifer* | Single | Cybaeus spider associated circular virus 1 |
| 2017 | Victoria BC | *Cybaeus signifer* | Single | Cybaeus spider associated circular virus 2 |
| 2017 | Victoria BC | *Cybaeus signifer* | Single | Cybaeus spider associated circular molecule 1 |
| 2017 | Victoria BC | *Steatoda grossa* (False black widow spider) | Single | False black widow spider associated circular virus 1 |
| 2017 | Victoria BC | *Eratigena duellica* (Giant house spider) | Single | Giant house spider associated circular virus 1 |
| 2017 | Victoria BC | *Eratigena duellica* (Giant house spider) | Single | Giant house spider associated circular virus 2 |
| 2017 | Victoria BC | *Eratigena duellica* (Giant house spider) | Single | Giant house spider associated circular virus 3 |
| 2017 | Victoria BC | *Eratigena duellica* (Giant house spider) | Single (2) | Giant house spider associated circular virus 4 |
| 2014 | Puerto Rico | *Nephila* sp. (Golden silk orbweaver)* | Single | Golden silk orbweaver associated circular virus 1 |
| 2017 | FL USA | *Leucauge argyra* (Longjawed orbweaver)* | Single | Longjawed orbweaver circular virus 1 |
| 2014 | Puerto Rico | *Leucauge argyra* (Longjawed orbweaver)* | Single | Longjawed orbweaver circular virus 2 |
| 2017 | Victoria BC | *Pimoa altioculata* (Pimoid spider)* | Single | Pimoid spider associated circular virus 1 |
| 2017 | Victoria BC | *Pimoa altioculata* (Pimoid spider)* | Single | Pimoid spider associated circular virus 2 |
| | | | | Pimoid spider associated circular molecule 1 |
| 2017 | Victoria BC | *Neriere litigiosa* (Sierra dome spider)* | Single | Sierra dome spider associated circular virus 1 |
| 2017 | Victoria BC | *Neriere litigiosa* (Sierra dome spider)* | Single | Sierra dome spider associated circular virus 2 |
| 2017 | Victoria BC | *Cybaeidae* (Soft spider) | Single | Soft spider associated circular virus 1 |
| 2017 | Victoria BC | *Cybaeus signifer* | Single | Spider associated circular virus 1 |
| 2017 | Victoria BC | *Segestria pacifica* (Tubeweb spider) | Single | Spider associated circular virus 1 |
| 2017 | Victoria BC | *Eratigena atrica* (Giant house spider) | Single | Spider associated circular virus 1 |
| 2017 | Victoria BC | *Parasteatoda tepidariorum* (Common house spider) | Single | Spider associated circular virus 2 |
| 2017 | Victoria BC | *Segestria pacifica* (Tubeweb spider)* | Single | Spider associated circular virus 3 |
| 2014 | FL USA | *Gasteracantha cancriformis* (Spinybacked orbweaver)* | Single | Spinybacked orbweaver circular virus 1 |
| 2017 | FL USA | *Gasteracantha cancriformis* (Spinybacked orbweaver)* | Single | Spinybacked orbweaver circular virus 1 |

(Continued)

| Year | Location[1] | Species name (common name)[2] | Samples[3] | Identified genomes |
|------|-------------|-------------------------------|------------|--------------------|
| 2017 | FL USA | *Gasteracantha cancriformis* (Spinybacked orbweaver)* | Single | Spinybacked orbweaver circular virus 2 |
| 2017 | FL USA | *Cyrtophora* sp. (Tentweb spider) | Single | Tentweb spider associated circular virus 1 |
| 2017 | Victoria BC | *Segestria pacifica* (Tubeweb spider)* | Single | Tubeweb spider associated circular virus 1 |
| 2017 | Victoria BC | *Dysdera crocata* (Woodlouse hunter spider) | Single | Woodlouse hunter spider associated circular virus 1 |
| 2015 | Kenya | *Odontotermes* sp. (Fungus-farming termite) | Pool | Termite associated circular virus 1 |
| | | | | Termite associated circular virus 3 |
| | | | | Termite associated circular virus 4 |
| 2015 | Kenya | *Odontotermes* sp. (Fungus-farming termite) | Pool (2) | Termite associated circular virus 2 |

Notes:
[1] Location abbreviations: FL, Florida; NH, New Hampshire; St. Barts, Saint Barthelemy; Dom. Republic, Dominican Republic; BC, British Columbia; Store, Carolina Biological Supply.
[2] Many specimens were taxonomically identified by sample providers. However, some specimens were identified through DNA barcoding and are indicated with an asterisk (*).
[3] Samples processed as individuals (Single) or pools (Pool) composed of up to 10 individuals from the same species are distinguished. Some CRESS DNA genomes were recovered from multiple individuals or pools (the number of samples that independently resulted in the identification of a given genome is specified within parenthesis). Although some genomes represent the same viral species, genomes sharing less than 100% genome-wide pairwise identity that were recovered from independent samples were submitted to GenBank and assigned individual accession numbers (see Table 3).

(Millipore, Burlington, MA, USA) and nucleic acids were extracted from 200 µl of filtrate using the QIAamp MinElute Virus Spin Kit (Qiagen, Hilden, Germany).

DNA barcoding was performed to identify any arthropods positive for CRESS DNA viruses that were not taxonomically identified by experts. For this purpose, DNA was extracted from tissue samples using the Quick-DNA Tissue/Insect Kit (Zymo Research, Irvine, CA, USA) following the manufacturer's instructions. The mitochondrial cytochrome oxidase I (COI) gene was then amplified through polymerase chain reaction (PCR) using the universal COI primers LCO1490 (5′GGTCAACAAATCATAAAGATA TTGG3′) and HCO2198 (5′TAAACTTCAGGGTGACCAAAAAATCA3′) (*Folmer et al., 1994*). A total of 50 µl PCR reactions contained the following: 1.5 mM $MgCl_2$, 1× Apex $NH_4$ Buffer, 0.5 µM primer LCO1490, 0.5 µM primer HCO2198, 5% DMSO, 1 µg/µl BSA, 1 U Apex Red Taq DNA Polymerase (Genesee Scientific, San Diego, CA, USA), and 3 µl of template DNA. Thermocycling conditions consisted of an initial denaturation at 95 °C for 2 min, followed by 35 cycles of 94 °C for 1 min, 48 °C for 1 min incrementally decreasing the temperature by 0.1 °C each cycle, and 72 °C for 1 min, with a final extension at 72 °C for 7 min. Mitochondrial COI gene PCR products were commercially sequenced using LCO1490 and HCO2198 primers. Sequences were compared against GenBank through BLASTn searches. Sequences sharing >95% identity with sequences in the database were classified to species, whereas sequences with nucleotide identities below this threshold were classified at the genus level.

## Detection of CRESS DNA viral genomes and genome completion

Small circular templates, such as CRESS DNA genomes, were enriched by amplifying DNA extracts through rolling circle amplification (RCA) using the Illustra TempliPhi Amplification Kit (GE Healthcare, Chicago, IL, USA) (*Haible, Kober & Jeske, 2006*; *Kim & Bae, 2011*). RCA products were digested using a suite of six-cutter FastDigest restriction enzymes (Thermo Fisher Scientific, Waltham, MA, USA), including *Bam*HI,

*Eco*RV, *Pdm*I, *Hind*III, *Kpn*I, *Pst*I, *Xho*I, *Sma*I, *Bgl*II, *Eco*RI, *Xba*I, and *Nco*I. Three microliters of RCA product from each sample were digested with each enzyme in separate reactions following the manufacturer's instructions to obtain complete, unit-length genomes. Products of the restriction digests were resolved on an agarose gel and fragments ranging in size from 1 to 4 kb were excised and purified using the Zymoclean Gel DNA Recovery Kit (Zymo Research, Irvine, CA, USA) for cloning. Since CRESS DNA viruses have been identified as contaminants in commercial spin columns used for nucleic acid extractions (*Krupovic et al., 2015*), negative controls containing SM Buffer alone were processed alongside samples from DNA extractions through restriction enzyme digests and, whenever applicable, PCR (see below).

In most cases, products resulting from blunt-cutting enzyme digestions were cloned into the pJET1.2 vector using the CloneJET PCR Cloning kit (Thermo Fisher Scientific, Waltham, MA, USA), whereas products resulting from enzymes producing sticky ends were cloned using pGEM-3Zf(+) vectors (Promega, Madison, WI, USA) pre-digested with the appropriate enzyme. However, if there were difficulties cloning into pre-digested pGEM-3Zf(+) vectors, sticky-end digestion products were cloned into the pJET1.2 vector following the manufacturer's sticky-end cloning protocol. Cloned digest products were then Sanger sequenced using vector primers. If these preliminary sequences showed significant similarities to CRESS DNA viral sequences based on BLASTn or BLASTx searches (*e*-value < 0.001), complete genome sequences were obtained through primer walking of cloned unit-length genomes or through inverse PCR using back-to-back primers designed from preliminary sequences. For the latter, PCR products were cloned using the CloneJET PCR Cloning kit and Sanger sequenced using vector primers and primer walking.

## CRESS DNA genome sequence analyses

Circular Rep-encoding ssDNA genome sequences were assembled in Geneious version R7 (Biomatters, Auckland, New Zealand) with default parameters for de novo assemblies. Major, non-overlapping ORFs >100 amino acids were identified and annotated using SeqBuilder from the Lasergene software package version 11.2.1 (DNASTAR, Madison, WI, USA) using the standard genetic code. Partial genes or genes that seemed interrupted were screened for potential introns using GENSCAN (*Burge & Karlin, 1997*). Genomes that did not contain a putative capsid-encoding ORF, based on BLASTx (*Altschul et al., 1990*) or remote protein homology searches using HHpred (*Söding, Biegert & Lupas, 2005*), were further investigated by looking at intrinsically disorder protein (IDP) profiles of non-Rep encoding ORFs using the neural network based VL3 disorder predictor on DisProt (*Sickmeier et al., 2007*). If non-Rep encoding ORFs contained a high proportion of disordered residues within the first 100 amino acids, they were considered putative capsid proteins (*Rosario et al., 2015a*). The potential *ori* for each genome was identified by locating the canonical nonanucleotide motif "NANTATTAC" observed in most CRESS DNA genomes (*Rosario, Duffy & Breitbart, 2012*), or similar sequences (*Krupovic et al., 2016*; *Varsani & Krupovic, 2018*), and evaluating if the identified nonamer was found at the apex of a predicted stem-loop structure using the Mfold web server (*Zuker, 2003*). Genome-wide and

Rep amino acid sequence pairwise identities (PIs) were calculated using the Sequence Demarcation Tool version 1.2 (*Muhire, Varsani & Martin, 2014*) to evaluate taxonomic relationships among CRESS DNA genomes identified in this study and those found in GenBank.

## Phylogenetic analyses

To evaluate how the novel CRESS DNA sequences identified in this study compared to previously reported CRESS DNA viral genomes, we constructed a phylogenetic tree from Rep amino acid sequences recovered from a wide array of organisms. For this purpose, Rep sequences were downloaded from GenBank in April 2018. These sequences included Reps from members of six established CRESS DNA viral families, including *Geminiviridae* (nine genera), *Nanoviridae* (two genera), *Circoviridae* (two genera), *Genomoviridae* (nine genera), *Bacilladnaviridae* (four genera), and *Smacoviridae* (six genera), as well as satellite molecules from the *Alphasatellitidae* (11 genera) and other CRESS DNA viral genomes that remain unclassified. To reduce the number of sequences in the analysis while still being able to assess diversity, Rep sequences representing established taxonomic groups were clustered based on a 70% amino acid identity cut off using CD HIT (*Fu et al., 2012*). However, if more than 20 sequences remained after clustering for a given group, sequences were clustered using a 50% identity cut off. All sequences outside of the established CRESS DNA families were grouped by organism (e.g., rodent associated sequences) and sequences within each group were clustered using a 70% amino acid identity cut off. In addition to sequences representing exogenous viruses and replicons, CRESS DNA-like endogenous viral sequences (CEVs) reported from various organisms were included (*Dennis et al., in press*, 2018; *Liu et al., 2011*). Selected CEVs did not contain any early stop codons or frameshifts. The final dataset contained 489 Rep sequences.

An alignment was performed using MUSCLE (*Edgar, 2004*) as implemented in MEGA7 (*Kumar, Stecher & Tamura, 2016*) and manually edited by inspecting and aligning sequences based on the presence of conserved RCR and superfamily 3 (SF3) helicase motifs (*Kazlauskas et al., 2017*; *Rosario et al., 2017*; *Varsani & Krupovic, 2017*, *2018*). The alignment was trimmed close to the RCR motif I and helicase arginine finger motifs and final aligned sequences, including CEVs, were at least 200 amino acids in length (Data S1). The alignment was used to construct an unrooted approximately-maximum-likelihood (ML) phylogenetic tree using FastTree 2 (*Price, Dehal & Arkin, 2010*) with default parameters. The phylogenetic tree was edited using TreeGraph 2 (*Stöver & Müller, 2010*) to collapse branches with support below a given threshold of Shimodaira–Hasegawa-like support and FigTree (http://tree.bio.ed.ac.uk/software/figtree/) was used for tree visualization and editing. The same alignment and tree editing strategies were used for all phylogenetic trees presented in this study.

Circular Rep-encoding ssDNA genome sequences representing species from established CRESS DNA viral groups were further investigated for genera and/or species level classification assignment. All species level assignments were based on current species demarcation criteria for CRESS DNA groups (Table 2). However, some genomes with similarities to members of the *Genomoviridae* and *Circoviridae* families seemed to

**Table 2 Taxonomic classification framework for established CRESS DNA viral groups.**

| Family | Genome-wide pairwise identity[1] | Species demarcation criteria[2] | Reference |
|---|---|---|---|
| *Alphasatellitidae* | 54% | *Geminialphasatellitinae*, 88% | *Briddon et al. (2018)* |
| | | *Nanoalphasatellitinae*, 80% | *Briddon et al. (2018)* |
| *Bacilladnaviridae* | Not reported | 75%* | *Kazlauskas et al. (2017)* |
| *Circoviridae* | 55% | 80% | *Rosario et al. (2017)* |
| *Geminiviridae* | 54% | *Becurtovirus*, 80% | *Varsani et al. (2014b)* |
| | | *Begomovirus*, 91% | *Brown et al. (2015)* |
| | | *Capulavirus*, 78% | *Varsani et al. (2017)* |
| | | *Curtovirus*, 77% | *Varsani et al. (2014a)* |
| | | *Glabovirus*, 80% | *Varsani et al. (2017)* |
| | | *Mastrevirus*, 78% | *Muhire, Varsani & Martin (2014)* |
| | | *Eragrovirus*, not reported | |
| | | *Turncurtovirus*, not reported | |
| | | *Topocuvirus*, not reported | |
| *Genomoviridae* | 53% | 78% | *Varsani & Krupovic (2017)* |
| *Nanoviridae* | Not reported | 75% | *Lefkowitz et al. (2018)* |
| *Smacoviridae* | 55% | 77% | *Varsani & Krupovic (2018)* |

Notes:
[1] Refers to the lower limit of genome-wide pairwise identities (PIs) among members of a given viral family.
[2] With the exception of the family *Bacilladnaviridae*, the species demarcation criteria (SDC) is based on genome-wide PIs. The SDC may vary by subfamily (*Alphasatellitidae*) or genus (*Geminiviridae*) within a given family.
* The SDC for the *Bacilladnaviridae* is based on amino acid sequence PI of the replication-associated protein.

fall outside existing genera; thus, further phylogenetic analyses were undertaken to determine the assignment of these CRESS DNA viruses as putative members of these families. Rep sequences representing members from each of the nine *Genomoviridae* genera and closely related viruses, including geminiviruses, were aligned and a ML phylogenetic tree was constructed using PhyML (*Guindon et al., 2010*) with the LG+G+I substitution model. The ML tree was then rooted with viral sequences from members of the *Geminiviridae* (*Varsani & Krupovic, 2017*). For the family *Circoviridae*, Rep sequences representing members from the *Circovirus* and *Cyclovirus* genera as well as closely related sequences and CEVs falling within this family were aligned. A midpoint rooted ML phylogenetic tree was then constructed using PhyML with automatic selection of substitution model through the Smart Model Selection using the Akaike Information Criterion (*Lefort, Longueville & Gascuel, 2017*).

# RESULTS

## CRESS DNA viruses identified in all three major lineages of terrestrial arthropods

More than 500 specimens representing a diversity of terrestrial arthropods were analyzed for the presence of CRESS DNA viruses through RCA followed by restriction enzyme digestion and cloning. Our efforts resulted in the detection of 44 unique (<80% genome-wide PI) CRESS DNA genomes (Table 3). Consistent with known CRESS DNA genomes, the genomes are small in size (<3.5 kb) and contain a putative *ori* marked

**Table 3 CRESS DNA genome information, accession numbers, and taxonomic groups identified in this study.**

| Accession number(s) | Genome[1] | Taxonomic affiliation[2] | Genome size (nt) | Nonanucleotide motif (type)[3] | BLAST match source (accession number)[4] | Identity (%)[5] Rep | Genome |
|---|---|---|---|---|---|---|---|
| MH545511–MH545513 | Arboreal ant associated circular virus 1 | *Circoviridae* | 1769 | TAGTATTAC (II) | Bat feces (KM382269) | 73* | 67 |
| MH545514 | Fly associated circular virus 1 | *Circoviridae* | 1722 | TAGTATTAC (II) | Cockroach (JX569794) | 89 | 85 |
| MH545516 | Soft spider associated circular virus 1 | *Circoviridae* | 1937 | TAGTATTAC (II) | Shrew feces (AB937987) | 63* | 59 |
| MH545515 | Spinybacked orbweaver circular virus 2 | *Circoviridae* | 1707 | TAGTATTAC (II) | Dragonfly (JX185424) | 99* | 99 |
| MH545522 | Cybaeus spider associated circular virus 1 | Circularisvirus | 1991 | TAATACTAC (V) | Dragonfly (JX185415) | 61 | 60 |
| MH545520 | Golden silk orbweaver associated circular virus 1 | Circularisvirus | 2054 | CAGTATTAC (V) | Dragonfly (JX185415) | 63 | 60 |
| MH545521 | Longjawed orbweaver circular virus 1 | Circularisvirus | 1905 | CATTATTAC (V) | Dragonfly (JX185415) | 60 | 62 |
| MH545518, MH545519 | Spinybacked orbweaver circular virus 1 | Circularisvirus | 1995 | CAGTATTAC (V) | Dragonfly (JX185415) | 63 | 64 |
| MH545523 | Fire ant associated circular virus 1 | Crucivirus | 3226 | TATGTGTAA (IV) | Wastewater (KY487859) | 61 | 55 |
| MH545497 | Bark beetle associated circular virus 1 | *Genomoviridae* | 2237 | TAATATTAT (II) | Dragonfly (JX185429) | 96* | 92 |
| MH545507 | Cybaeus spider associated circular virus 2 | *Genomoviridae* | 2344 | TAATATTAT (II) | Whitefly (KY230625) | 67* | 61 |
| MH545498 | Fly associated circular virus 2 | *Genomoviridae* | 2207 | TAACATTGT (II) | Pig feces (KY214433) | 99* | 99 |
| MH545509 | Giant house spider associated circular virus 1 | *Genomoviridae* | 2093 | TAATATTAT (II) | Llama feces (KT862245) | 73* | 67 |
| MH545499 | Grasshopper associated circular virus 1 | *Genomoviridae* | 2309 | TAACACTGT (II) | Bat feces (KT732803) | 62* | 64 |
| MH545500 | Pimoid spider associated circular molecule 1[A] | *Genomoviridae* | 1662 | TAATGTTAT (II) | Llama feces (KT862245) | 69* | 68 |
| MH545508 | Pimoid spider associated circular virus 1 | *Genomoviridae* | 2240 | TAATATTAT (II) | Sewage (KJ547640) | 100* | 99 |
| MH545510 | Sierra dome spider associated circular virus 1 | *Genomoviridae* | 2232 | TAATATTAT (II) | Bird feces (KF371636) | 67* | 64 |
| MH545503–MH545505 | Spider associated circular virus 1 | *Genomoviridae* | 2214-2216 | TAATACTAT (II) | Cow feces (KT862253) | 84* | 74 |
| MH545506 | Spider associated circular virus 2 | *Genomoviridae* | 2204 | TAATACTAT (II) | Cow feces (KT862253) | 85* | 71 |
| MH545502, MG917675 | Termite associated circular virus 2 | *Genomoviridae* | 2222-2226 | TAATATTAT (II) | Thrips (KY308268) | 74* | 68 |
| MH545501 | Tubeweb spider associated circular virus 1 | *Genomoviridae* | 2174 | TAACACTGT (II) | Thrips (KY308270) | 63* | 61 |
| MH545524 | Fly associated circular virus 3 | *Smacoviridae* | 2537 | TAGTGTTAC (IV) | Macaques feces (KU043428) | 83 | 89 |
| MH545525 | Fly associated circular virus 4 | *Smacoviridae* | 2546 | TAGTGTTAC (IV) | Chimpanzee feces (GQ351275) | 57 | 61 |
| MH545526 | Cricket associated circular virus 1 | Volvovirus | 2516 | TAGTATTAC (II) | Cricket (KC794539) | 100 | 99 |
| MH545538 | Common house spider circular molecule 1[B] | Unclassified | 1833 | TATTATTAC (V) | Giant panda feces (MF327573) | 62 | 63 |
| MH545543 | Cybaeus spider associated circular molecule 1[B] | Unclassified | 1989 | TAGCACTAA (VIII) | Peatland (KX388505)[#] | 58* | n/a |
| MH545542 | False black widow spider associated circular virus 1 | Unclassified | 2199 | TAGTATTAC (I) | Reclaimed water (NC_013023) | 61 | 62 |
| MH545517 | Fly associated circular virus 5 | Unclassified | 1997 | TAGTATTAC (II) | Bat feces (KT732825) | 97 | 93 |
| MH545530 | Fly associated circular virus 6 | Unclassified | 2103 | TAGTATTAC (IV) | Wastewater (KY487991)[#] | 61 | 59 |

| Accession number(s) | Genome[1] | Taxonomic affiliation[2] | Genome size (nt) | Nonanucleotide motif (type)[3] | BLAST match source (accession number)[4] | Identity (%)[5] Rep | Genome |
|---|---|---|---|---|---|---|---|
| MH545531 | Fly associated circular virus 7 | Unclassified | 2010 | TAGTATTAC (IV) | Wastewater (KY487991)# | 65 | 66 |
| MH545536 | Giant house spider associated circular virus 2 | Unclassified | 2040 | TAGTATTAC (V) | Sphaeriid clam (KP153475) | 68 | 64 |
| MH545537 | Giant house spider associated circular virus 3 | Unclassified | 2290 | TATTATTAC (I) | Amphipod (KC248416) | 61 | 59 |
| MH545541 | Giant house spider associated circular virus 4 | Unclassified | 2494 | TAATATTAC (IV) | Wastewater (KY487963) | 62 | 60 |
| MH545529 | Longjawed orbweaver circular virus 2 | Unclassified | 2321 | CAGTATTAC (VI) | Damselfly (KM598408) | 58 | 57 |
| MH545532 | Millipede associated circular virus 1 | Unclassified | 1987 | TAGTATTAC (II) | Estuarine snail (NC_026646) | 59 | 58 |
| MH545534 | Pimoid spider associated circular virus 2 | Unclassified | 2125 | TAGTATTAC (I) | Bat (KJ641721)# | 62 | 60 |
| MH545535 | Sierra dome spider associated circular virus 2 | Unclassified | 1860 | TAGTATTAC (V) | Giant panda feces (NC_035196) | 57 | 57 |
| MH545539 | Spider associated circular virus 3 | Unclassified | 1889 | CAACCACTC (I) | Ice shelf pond (NC_024478) | 57 | 57 |
| MH545533 | Tentweb spider associated circular virus 1 | Unclassified | 2127 | TAGTATTAC (II) | Dragonfly larvae (KF738884) | 62 | 60 |
| MG917674 | Termite associated circular virus 1 | Unclassified | 2155 | TAATATTAC (II) | Chicken feces (KY056250) | 61* | 55 |
| MG917676 | Termite associated circular virus 3 | Unclassified | 2220 | TAATGTTAC (II) | Shrub (KT214387) | 57* | 56 |
| MG917677 | Termite associated circular virus 4 | Unclassified | 2152 | TAATGTTAC (II) | Tomato (NC_036591) | 58* | 57 |
| MH545527, MH545528 | Water beetle associated circular virus 1 | Unclassified | 2244 | CAGTATTAC (II) | Ice shelf pond (NC_024478) | 56 | 57 |
| MH545540 | Woodlouse hunter spider associated circular virus 1 | Unclassified | 2176 | TAATAGTAG (II) | Amphipod (KC248416) | 57* | 58 |

**Notes:**
[1] A few genomes were not considered viral and were labelled as "molecules" for the following reasons: (A) capsid-encoding open reading frame (ORF) seemed truncated or (B) genome only contained a single major ORF.

[2] Groups that do not represent established taxonomic groups by ICTV are non-italicized, including Circularisvirus, Crucivirus, and Volvovirus.

[3] Most nonamers were located at the apex of a predicted hairpin structure, with the exception of a circular molecule identified with the symbol (^). Genome organizations, using the specified nonamer as a reference, are indicated within parenthesis according to genotypes discussed by Rosario, Duffy & Breitbart, 2012.

[4] Best BLAST matches for identified CRESS DNA genomes. Some of the most closely related viruses to CRESS DNA viruses and replicons identified here, based on BLAST searches, contain a different genome organization and are indicated with the symbol (#).

[5] Pairwise identities (PIs) between identified CRESS DNA genomes and their best BLAST match. Nucleotide PIs between replication-associated proteins (Rep) were calculated based on predicted spliced coding regions. Genomes containing Rep-encoding ORFs interrupted by an intron are marked with the symbol (*).

by a conserved nonanucleotide motif at the apex of predicted stem-loop structure. Based on the position of the *ori* relative to major ORFs, six out of the eight described CRESS DNA genome organizations were detected. (*Rosario, Duffy & Breitbart, 2012*). In addition to an identifiable Rep-encoding ORF, most (*n* = 34) of these CRESS DNA genomes encode a putative capsid protein based on similarities to capsid proteins found in public databases. A total of 10 of the detected genomes do not encode an ORF with significant matches to known capsid proteins; however, the non-Rep encoding ORFs in eight of these genomes have similar IDP profiles to those seen in other CRESS DNA viruses, suggesting they encode a putative capsid protein (Data S2) (*Rosario et al., 2015a*). The three genomes for which a capsid-encoding ORF could not be identified were named as "circular molecules" to distinguish these replicons from bona fide CRESS DNA viral genomes.

Half of the CRESS DNA genomes described here were identified in spiders (Class Arachnida; Order Araneae), despite the fact that >70% of the samples processed in this study were insects (Class Insecta; data not shown). The high prevalence of CRESS DNA genomes in spiders is even more striking considering that most spiders were processed individually, as opposed to many of the insect species for which multiple individuals were pooled (Table 1). To our knowledge, these genomes represent the first exogenous CRESS DNA viruses reported from spiders. CRESS DNA viruses were also widely detected in insects with viral genomes retrieved from specimens representing five orders, including ants (Hymenoptera), beetles (Coleoptera), flies (Diptera), grasshoppers and crickets (Orthoptera), and termites (Blattodea). In addition, we detected a genome from a millipede (Class Diplopoda) representing the first CRESS DNA virus associated with a member of the Subphylum Myriapoda. The low number of CRESS DNA viruses identified from members of the Myriapoda may be a consequence of uneven sampling since only seven specimens from this group were processed. Similarly, groups for which no CRESS DNA viruses were identified, including Odonata, Dermaptera, Lepidoptera, Ephemeroptera, and Chilopoda, had low sample numbers (<10) (data not shown).

## Terrestrial arthropods harbor diverse novel CRESS DNA viruses

Over half of the genomes (55%) identified in this study shared <70% genome-wide PI with previously reported sequences (Table 3) and could not be assigned to an existing CRESS DNA group. Phylogenetic analysis of Rep amino acid sequences retrieved from a wide array of organisms illustrated the wide phylogenetic distribution of the arthropod CRESS DNA viruses and replicons identified here (Fig. 1). Some of the arthropod-associated CRESS DNA genomes falling outside of established taxonomic groups were most closely related to isolates that have not been assigned to either genera or families. We identified four genomes from spiders that were most closely related to a circularisvirus reported from dragonflies (*Rosario et al., 2012*). Phylogenetic analysis revealed other circularisvirus-like genomes retrieved from dragonflies (accession KM598396, *Dayaram et al., 2015b*) and bat feces (accession KT732823, *Male et al., 2016*) (Fig. 1). All of these circularisvirus-like genomes contained similar genomic features, including unisense organization, similar size (~1.9–2 kb), and a putative *ori* on the

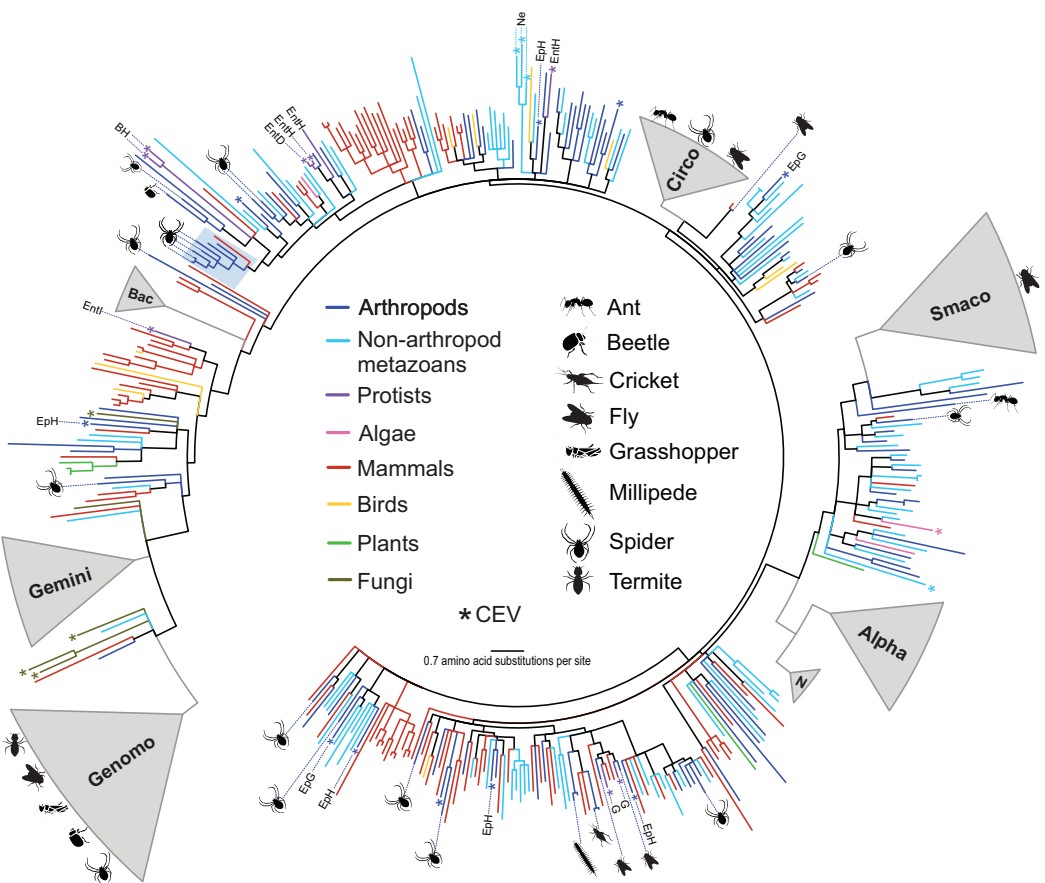

**Figure 1 Approximately maximum likelihood phylogenetic tree of replication-associated protein (Rep) amino acid sequences representing CRESS DNA viruses, replicons, and CRESS DNA-like endogenous viral (CEV) elements recovered from various organisms.** Branch colors distinguish sequences associated with various types of organisms. Clades containing Rep sequences falling within established CRESS DNA viral groups including the *Genomoviridae* (Genomo), *Geminiviridae* (Gemini), *Bacilladnaviridae* (Bac), *Circoviridae* (Circo), *Smacoviridae* (Smaco), *Nanoviridae* (N) and *Alphasatellitidae* (Alpha) were merged and are highlighted in gray. The new circularisvirus clade is highlighted with a light blue rectangle. The asterisk symbol indicates branches representing CEVs. Branches representing CEVs identified in *Ephydra* spp. (dipteran), including *E. gracilis* (EpG) and *E. hians* (EpH), and nematodes (Ne) are specified. CEVs identified in protists are also specified, including *Entamoeba* (Ent), *Giardia intestinalis* (G) and *Blastocystis hominis* (BH). *Entamoeba* species are further distinguished, namely *E. invadens* (EntI), *E. histolytica* (EntH), and *E. dispar* (EntD). Reps identified in this study are highlighted with schematics of terrestrial arthropods showing their source and broad phylogenetic distribution. Branches with <80% Shimodaira–Hasegawa (SH)-like support were collapsed. Arthropod silhouettes credit: Shutterstock vector library at https://www.shutterstock.com.

Rep-encoding strand (Table 3). In addition, circularisvirus-like genomes shared >57% genome-wide PIs among each other, which is similar to genome-wide PIs reported for established CRESS DNA viral families (Table 2). Conserved circularisvirus genomic characteristics and genome-wide PIs may grant the formation of a new group of CRESS DNA viruses.

Fire ant associated circular virus 1 (FaACV-1) has features characteristic of the crucivirus group, members of which have been mainly reported from environmental

samples (*Diemer & Stedman, 2012*; *Krupovic et al., 2015*; *McDaniel et al., 2014*; *Quaiser et al., 2016*; *Roux et al., 2013*; *Steel et al., 2016*). Namely, the FaCV-1 genome contains a Rep-encoding ORF most similar to that of CRESS DNA viruses and a putative capsid protein with significant similarities to capsid proteins of ssRNA viruses from the family *Tombusviridae*. FaCV-1 is most closely related to a crucivirus isolated from wastewater (Table 3). Both FaCV-1 and the wastewater associated crucivirus have two major ORFs arranged in ambisense orientation and share 55% genome-wide PI, suggesting that these genomes may belong to the same viral family. However, currently there is no classification framework for cruciviruses. In addition, we identified a molecule, cybaeus spider associated circular molecule 1, containing a single ORF most similar to a Rep-encoding ORF from a crucivirus identified from peatland (*Quaiser et al., 2016*). The findings presented here suggest that cruciviruses circulate in insects and may be associated with both terrestrial and aquatic (*Bistolas et al., 2017*; *Hewson et al., 2013b*) arthropods.

Although the aim of this study was to identify CRESS DNA genomes, four novel circular molecules that did not encode a Rep were detected (Data S2). These included two molecules, leaf-footed bug associated circular molecule 1 (LfBACM-1) and Spider associated circular molecule 2 (SACM-2), that only contained a single major ORF encoding a putative capsid. The small genome size (<1.2 kb) of these molecules is reminiscent of capsid-encoding genomic segments from multipartite CRESS DNA viruses from the family *Nanoviridae* (*Gronenborn, 2004*). Indeed, LfBACM-1 is most similar to a genomic segment from a novel multicomponent CRESS DNA virus discovered in the feces of fruit-eating bats (*Male et al., 2016*). However, the SACM-2 putative capsid protein sequence is most similar to the capsid encoded by a presumably monopartite CRESS DNA virus discovered from a sewage oxidation pond (*Kraberger et al., 2015*). Surprisingly, the remaining two molecules, longjawed orbweaver circular molecule 1 (LjOrbCM-1) and giant house spider associated circular molecule 1 (GhSACM-1), encoded a protein most similar to the large T antigen (LT) encoded by polyomaviruses. The LjOrbCM-1 genome only contained the LT-encoding ORF, whereas GhSACM-1 encoded an additional major, non-overlapping ORF. However, the non-LT encoding ORF of GhSACM-1 was not predicted to encode a structural protein based on homology searches or IDP profiles. These four non-CRESS DNA molecules will not be discussed further, but these findings are noteworthy since they support studies describing capsid-encoding molecules potentially representing novel multipartite viruses associated with unsuspected organisms (*Male et al., 2016*) and the presence of episomal polyoma-like replicons in spiders (*Buck et al., 2016*).

## Terrestrial arthropods harbor a diversity of species representing new members of established CRESS DNA viral groups

The CRESS DNA genomes that could be assigned to previously reported taxa were dominated by members of the family *Genomoviridae*, which included genomes retrieved from spiders ($n = 7$), flies ($n = 2$), grasshoppers ($n = 1$), and termites ($n = 1$) (Table 3). Phylogenetic analysis based on the Rep indicated that the newly identified viruses belong to three genera (*Gemycircularvirus*, *Gemykibivirus*, and *Gemykolovirus*)

within the family *Genomoviridae* (Fig. 2; Fig. S1). The majority of arthropod-associated genomoviruses identified here belong to the genus *Gemycircularvirus*, which is the genus containing the highest number of species within the family (*Varsani & Krupovic, 2017*). Based on the species demarcation criteria of 78% genome-wide PI (Table 2), two of the seven identified gemycircularviruses represent new isolates from the classified species dragonfly associated gemycircularvirus 1 and sewage derived gemycircularvirus 4. The remaining five gemycircularviruses represent new species. Spider associated circular viruses (SACVs) 1 and 2 represent a new gemycircularvirus species that was identified in four species of spiders, with SACV-1 and -2 isolates sharing 79–98% genome-wide PI. In addition to gemycircularviruses, we identified two isolates, fly associated circular virus 2 (FlyACV-2) and cybaeus spider associated circular virus 2 (CySACV-2), representing members of the genus *Gemykibivirus*. CySACV-2 represents a novel gemykibivirus species, whereas FlyACV-2 is a variant (92% genome-wide PI) of an unclassified species currently represented by a gemykibivirus isolate reported from pig feces (*Nádia et al., 2017*). Lastly, grasshopper associated circular virus 1 and tubeweb spider associated circular virus 1, represent two new species of the genus *Gemykolovirus*.

In addition to viral genomes that clearly fall within the well-established family *Genomoviridae*, we identified three genomes from fungus-farming termites that belong to a group of unclassified viruses that appear to be intermediate between genomoviruses and geminiviruses (Fig. 2). In a brief report noting the prevalence of these termite associated circular viruses (TACVs) in African *Odontotermes* sp. mounds we indicated that these genomes were most similar to members of the *Genomoviridae* (*Kerr et al., 2018*). However, Rep phylogenetic analysis indicate that only TACV-2 belongs to the *Genomoviridae* (genus *Gemycirculovirus*), while TACV-1, -3, and -4 represent a new group of viruses. Furthermore, TACV-3 and -4 have top BLAST matches to geminiviruses (Table 3), but these genomes cluster closer to genomoviruses than geminiviruses (Fig. 2). There are a number of unclassified sequences retrieved from various environmental sources that fall in a similar phylogenetic position with TACV-1, -3, and -4, which may grant the formation of new taxonomic groups.

Members of the family *Circoviridae*, genus *Cyclovirus*, were detected in spiders ($n = 2$), flies ($n = 1$), and ants ($n = 1$) (Table 3). Based on the species demarcation criteria of 80% genome-wide PI (Table 2), two of these genomes, arboreal ant associated circular virus 1 (AaACV-1) and soft spider associated circular virus 1 (SoSACV-1), represent novel cyclovirus species. Fly associated circular virus 1 (FlyACV-1) and spinybacked orbweaver circular virus 2 (SpOrbCV-2) are new isolates of the classified species cockroach associated cyclovirus 1 (CroACV-1) and dragonfly associated cyclovirus 3 (DfACyV-3), respectively. While FlyACV-1 seems to be a divergent variant of the CroACV-1 species, sharing 85% genome-wide PI with this cyclovirus, SpOrbCV-2 shares 99% PI with DfACyV-3. Interestingly, DfACyV-3 was discovered from a dragonfly collected in the same region in FL, USA (*Rosario et al., 2012*) as SpOrbCV-2, indicating that this cyclovirus species has been circulating in the region for at least 7 years.

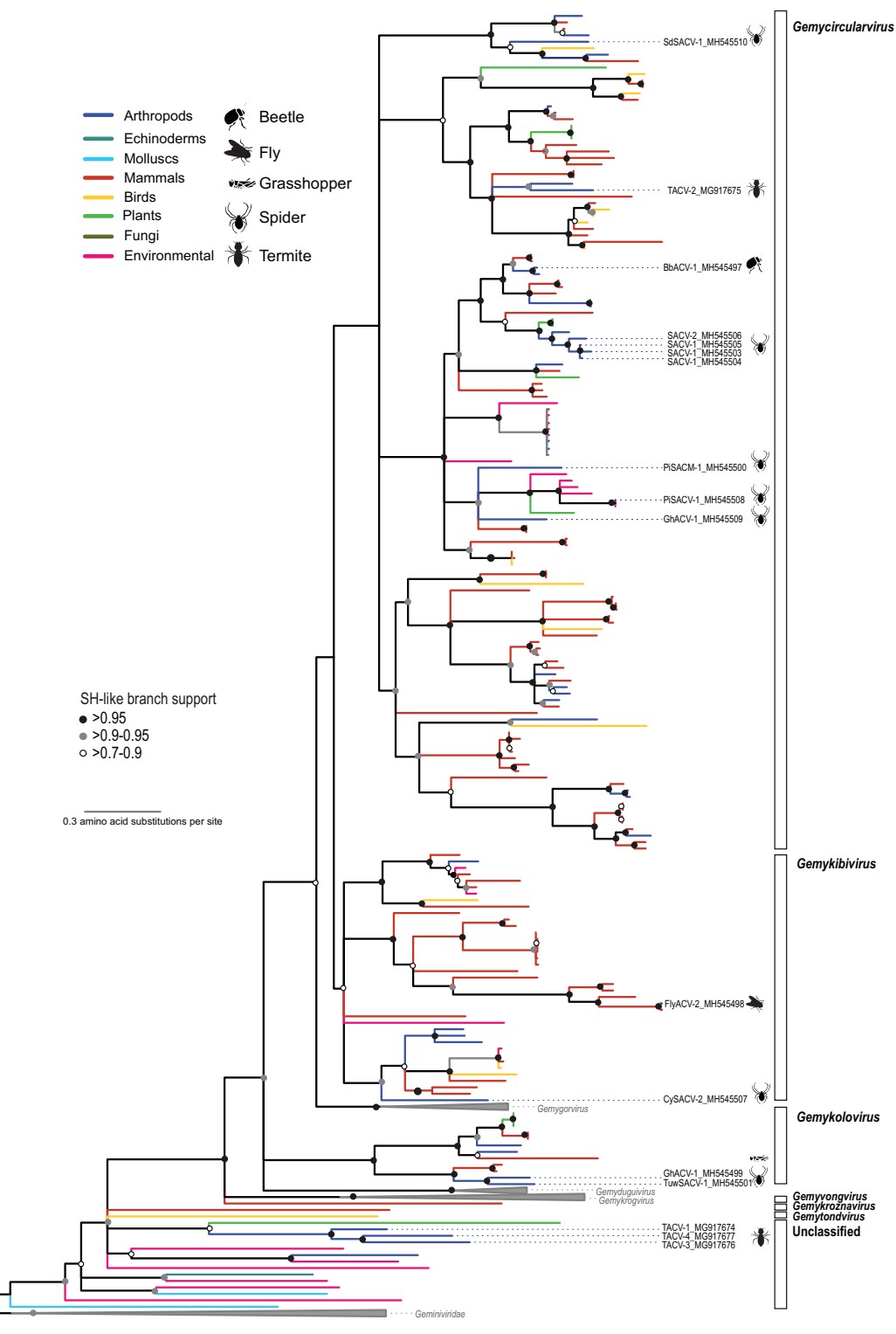

**Figure 2 Maximum likelihood phylogenetic tree of replication-associated protein (Rep) amino acid sequences representing members of the family *Genomoviridae* and related CRESS DNA viruses.** Branch colors distinguish sequences associated with various types of organisms and environmental sources. Bars on the right indicate clades representing genomovirus genera and unclassified sequences. Clades containing Rep sequences representing *Gemygorvirus*, *Gemyduguivirus*, and *Gemykrogvirus* species and members from the family *Geminiviridae*, which were used as an outgroup, were merged. Genomovirus Reps identified in this study are named and highlighted with schematics of terrestrial arthropods from which they were identified, including viruses associated with sierra dome spiders (SdSACV), pimoid spiders (PiSACV), tubeweb spiders (TuwSACV), grasshoppers (GhACV), and termites (TACV). Viruses identified in multiple species of spiders are identified as spider associated circular viruses (SACV). Branches with <70% Shimodaira–Hasegawa (SH)-like support were collapsed. A version of the tree containing source information and accession numbers for all the sequences included in the phylogenetic analysis is available in Fig. S1. Arthropod silhouettes credit: Shutterstock vector library at https://www.shutterstock.com.      

In addition to the four cycloviruses, we identified a genome, fly associated circular virus 5 (FlyACV-5), which is most closely related to cycloviruses and shares the genomic features characteristic of members in this genus, including genome organization, size, and putative *ori* (Table 3). Genome-wide PIs between FlyACV-5 and known cyclovirus species are within the accepted range for members of the *Circoviridae* (>55% genome-wide PIs) (*Rosario et al., 2017*). However, phylogenetic analysis of Rep sequences from members of the family *Circoviridae* did not support the placement of FlyACV-5 in either of the established genera for this family (Fig. 3). FlyACV-5 was most closely related to a CRESS DNA virus retrieved from bat feces, the Pacific flying fox feces associated circular DNA virus-8 (PfffACV-8, accession KT732825) (*Male et al., 2016*). Since both FlyACV-5 and PfffACV-8 have genomic features characteristic of the genus *Cyclovirus* and share genome-wide PIs >55% with members of this genus, these genomes may represent a novel group within the family *Circoviridae*. The phylogenetic analysis also revealed two cycloviruses, namely SoSACV-1 (accession MH545516) and Pacific flying fox feces associated circular DNA virus-2 (PfffACV-2, accession KT732786) that seem to be intermediate between circoviruses and cycloviruses (Fig. 3). However, at present, these genomes have been classified as cycloviruses based on their genome organization, which is a mirror image of that observed in circoviruses (*Rosario et al., 2017*).

Two new members of the family *Smacoviridae* were identified in flies (Table 3). Both isolates, fly associated circular viruses (FlyACV) -3 and -4, represent new species belonging to the genus *Porprismacovirus* based on the species demarcation criteria (Table 2) and genus demarcation threshold of 40% Rep amino acid sequence PI (*Varsani & Krupovic, 2018*). *Porprismacovirus* is by far the genus with the highest number of species in the family (*Varsani & Krupovic, 2018*). Although both FlyACV-3 and -4 represent new species, FlyACV-3 is closely related to an unclassified smacovirus isolated from macaque feces (*Kapusinszky et al., 2017*). Therefore, FlyACV-3 and the unclassified macaque associated smacovirus represent variants of the same *Porprismacovirus* species.

In addition to viruses most closely related to members of established CRESS DNA taxonomic groups, we identified an isolate representing a cricket-infecting virus that has not been classified. Cricket associated circular virus 1 (CrACV-1), identified in

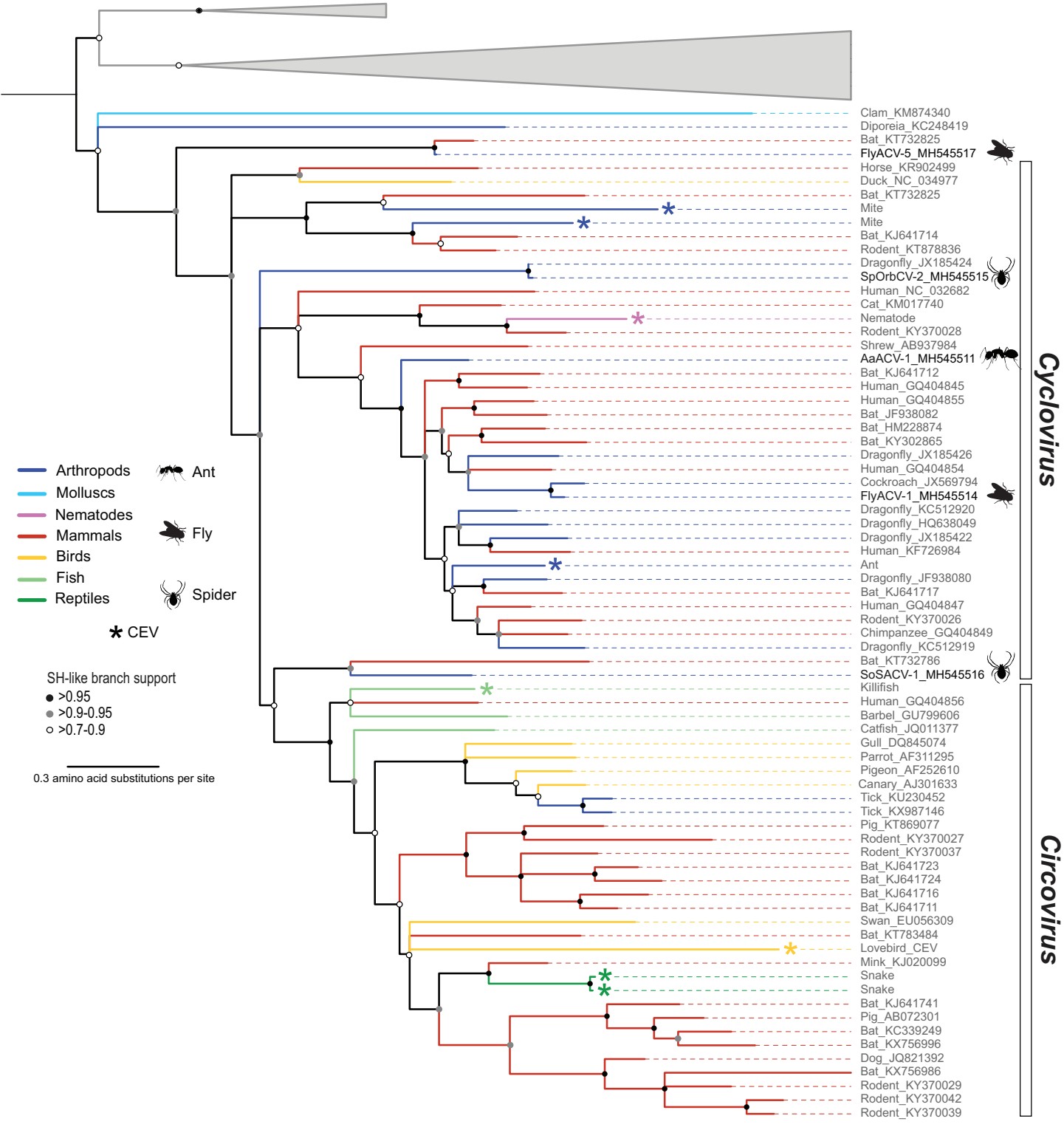

**Figure 3** Midpoint rooted maximum likelihood phylogenetic tree of selected replication-associated protein (Rep) amino acid sequences representing members of the family *Circoviridae* and related CRESS DNA viruses. Branch colors distinguish sequences associated with various invertebrate and vertebrate organisms. Bars on the right indicate clades representing the *Cyclovirus* and *Circovirus* genera. Rep sequences representing CRESS DNA-like endogenous viral (CEV) elements are highlighted with an asterisk symbol. Cyclovirus Reps identified in this study are highlighted with schematics of terrestrial arthropods and include viruses identified from flies (FlyACV), ants (AaACV), soft spiders (SoSACV) and spinyback orbweavers (SpOrbCV). Reps representing unclassified genome sequences forming non-*Circoviridae* clades used as outgroups were merged and are highlighted in gray (accessions: KX246259, KR528563, KM598407, KR528546, KM874290, KM874319, KM874343, KT945164). Branches with <70% Shimodaira–Hasegawa (SH)-like support were collapsed. Arthropod silhouettes credit: Shutterstock vector library at https://www.shutterstock.com.

a store-bought cricket, represents an isolate of *Achaeta domesticus* volvovirus (*Pham, Bergoin & Tijssen, 2013a*; *Pham et al., 2013b*). The four volvovirus genomes that have been reported to date, including CrACV-1, have been recovered from commercial crickets and share >99% genome-wide PI, thus representing a single viral species.

## Detection of a cyclovirus endogenous element in a non-arthropod invertebrate

Analysis of BLASTn matches for fly associated circular viruses (FlyACV) -6 and -7 in the GenBank non-redundant database revealed weak matches to nematodes. Although these initial BLAST matches were weak (query coverage < 15%), this prompted a search in the WormBase Parasite database (*Howe et al., 2017*) using the FlyACVs putative Rep sequences as queries. This search led to the detection of a previously unreported CEV from *Hymenolepis microstoma*, commonly known as rodent tapeworm. The *H. micostoma* CEV is embedded within a 251 kb genome scaffold (accession LN902886) (*Tsai et al., 2013*). The GenBank record for this genome scaffold noted two Rep-associated coding sequences (CDS) that were positioned next to each other. One of the CDS contains a near full-length Rep (accession CDS32196), whereas the second one (accession CDS32195) is interrupted at the SF3 Walker-A motif, at which point a partial capsid is encoded in the same reading frame. Inclusion of the near full-length Rep from *H. microstoma* in the phylogenetic analysis showed that this sequence falls within the *Cyclovirus* genus (Fig. 3). Although CEVs have been previously noted from parasitic helminths (*Liu et al., 2011*), this is the first cyclovirus CEV reported from nematodes. Notably, the *H. microstoma* CEV Rep sequence is most closely related to a cyclovirus sequence reported from rodents (accession KY370028). The putative endogenous capsid sequence is most similar to a cyclovirus reported from cat feces (*Zhang et al., 2014*), which is also closely related to the *H. microstoma* CEV and the rodent cyclovirus based on the Rep (Fig. 3). Therefore, both the CEV Rep and capsid sequences indicate that *H. microstoma* has been infected at some point by a cyclovirus. To our knowledge, this is the first evidence indicating that non-arthropod invertebrates serve as hosts for cycloviruses.

## DISCUSSION
### Most CRESS DNA viral diversity circulates among arthropods and other invertebrates

From the relatively small-scale survey presented here, it is clear that terrestrial arthropods harbor an extensive diversity of CRESS DNA viruses. Combining our results with previous

reports, exogenous CRESS DNA viruses have now been reported in organisms from four out of the five major branches of the Arthropod Tree of Life (*Giribet & Edgecombe, 2012*), including Euchelicerata (Class Arachnida) (*Kraberger et al., 2018*; *Wang et al., 2018*), Hexapoda (Class Insecta) (*Dayaram et al., 2013, 2015b*; *Kraberger et al., 2017*; *Padilla-Rodriguez, Rosario & Breitbart, 2013*; *Rosario et al., 2012*; *Tikhe & Husseneder, 2017*), Myriapoda (Class Diplopoda), and Crustacea (Classes Malacostraca, Maxillopoda, Copepoda, Branchiopoda) (*Bistolas et al., 2017*; *Dunlap et al., 2013*; *Hewson et al., 2013b*; *Rosario et al., 2015a*). To the best of our knowledge, no studies have specifically looked for CRESS DNA viruses within the remaining major arthropod branch, the Pycnogonida (sea spiders).

Spiders were identified as an unsuspected rich reservoir of CRESS DNA viral diversity, harboring most of the genomes from distinct viral groups identified in this survey. However, it should be noted that spiders are insectivores; thus, it is possible that the wide array of CRESS DNA viral diversity they contain is partially the result of accumulating CRESS DNA viruses from their insect prey. Similarly, a high diversity of CRESS DNA viruses has been reported from dragonflies, which are also top insect predators (*Dayaram et al., 2013*; *Rosario et al., 2012*). Since our methods might have recovered viruses from dietary content, it is possible that generalist arthropod predators may contain a broader range of CRESS DNA viruses than dietary specialists. Additionally, the discovery of some CRESS DNA viruses in multiple arthropod species may be due to overlapping diets. Phylogenetic analysis of the conserved Rep indicates that many of the diverse CRESS DNA viruses found within the terrestrial arthropods and other invertebrates fall outside established CRESS DNA viral families and do not form cohesive phylogenetic groups. This observation suggests that the CRESS DNA viral diversity associated with arthropods and other invertebrates has been grossly underestimated and that additional sampling of these groups would continue to expand the CRESS DNA virosphere. A more systematic survey targeting the same number of specimens from different taxonomic groups and representing a wider geographic distribution, rather than the opportunistic sampling effort shown here, may help elucidate which arthropod groups are hot spots for CRESS DNA viral diversity.

The diversity of CRESS DNA genomes identified in this study spans the entire CRESS DNA phylogenetic tree representing Reps recovered from a wide array of eukaryotic organisms (Fig. 1). Moreover, the CRESS DNA diversity falling outside established taxonomic groups that has been detected within arthropods and other invertebrates overwhelms the diversity reported from vertebrate organisms and plants, despite the fact that the latter groups have been heavily sampled. Members of the *Alphasatellitidae*, *Geminiviridae*, and *Nanoviridae* families were not detected in our survey, which did not include plant virus insect vectors. However, these plant-infecting CRESS DNA viruses and satellite molecules have long been known to circulate among hemipteran vectors (*Hogenhout et al., 2008*), which have been exploited to discover viral species found in a given area (*Ng et al., 2011*; *Rosario et al., 2015b, 2016*). Therefore, arthropod-associated viruses include members from five out of the six CRESS DNA viral families that have been identified as monophyletic (*Kazlauskas et al., 2017*). The remaining established

CRESS DNA family that was not identified in our survey, *Bacilladnaviridae*, includes viruses infecting unicellular algae and has only been reported from aquatic environments (*Kazlauskas et al., 2017*).

We identified novel viruses representing new members from the *Genomoviridae*, *Smacoviridae*, and *Circoviridae* families. Genomoviruses were the most diverse group, with 12 genomes recovered from spiders and insects from the orders Blattodea, Coleoptera, Diptera, and Orthoptera. The arthropod-associated genomoviruses represent at least three genera and highlight the ever growing diversity and wide distribution of this group of viruses (*Krupovic et al., 2016*). Two CRESS DNA viral sequences representing members of the *Smacoviridae*, which have been mainly recovered from feces from various mammals (*Ng et al., 2015*; *Steel et al., 2016*; *Varsani & Krupovic, 2018*), were recovered from blow flies (Diptera: Calliphoridae) collected in the Caribbean. The novel fly associated smacoviruses, FlyACV-3 and -4, were recovered from blow flies collected using a chicken carcass as bait (*Yusseff-Vanegas & Agnarsson, 2017*). Since blow flies are known to feed on feces and tissues from various vertebrates, including mammals, it is likely that FlyACV-3 and -4 represent dietary content. Nevertheless, the detection of fly associated smacoviruses and a smacovirus in dragonflies (*Dayaram et al., 2015b*) indicates that this group of viruses circulates within arthropods.

Members of the family *Circoviridae* present a unique distribution relative to all of the other established CRESS DNA viral groups infecting multicellular organisms. All of the *Circoviridae* members identified in this study represent the genus *Cyclovirus*. Although cycloviruses have been identified in both vertebrates and arthropods, to date, members of the genus *Circovirus* have mainly been reported from the former. Moreover, this observation is consistent with CEV searches (*Belyi, Levine & Skalka, 2010*; *Dennis et al., in press*; *Liu et al., 2011*) including a survey of more than 680 animal genomes, ~50% of which were invertebrates (*Dennis et al., 2018*). Viruses reported from ticks are the only arthropod-associated CRESS DNA viruses belonging to the genus *Circovirus* (*Tokarz et al., 2018*; *Wang et al., 2018*). Since ticks are hematophagous parasites that feed exclusively on the blood of birds and mammals (*Basu & Charles, 2017*), it is possible that tick associated circoviruses represent vertebrate-infecting viruses, in particular avian circoviruses (Fig. 3). Interestingly, bona fide circoviruses have not been reported from mosquitoes, a major group of blood-feeding arthropods of public health relevance. However, this might reflect the scarcity of mosquito DNA viromes reported to date. Despite these caveats, the available data suggests that cycloviruses circulate in a wide array of invertebrates and mammals, whereas circoviruses are mainly restricted to vertebrates and, perhaps, blood-feeding arthropod vectors.

In the Rep-based phylogeny, cycloviruses appear basal with respect to circoviruses (Fig. 3). Based on the higher diversity of cycloviruses described to date and the wider distribution of these viruses in both vertebrates and invertebrates, it is conceivable that cycloviruses are ancestral to circoviruses. We also note that there is a group of cycloviruses recovered from spiders and an insectivorous bat that seem intermediate between other cycloviruses and the circovirus clade (Fig. 3). Moreover, we detected viral sequences with cyclovirus genome organization from blow flies and an insectivorous bat that do not

fall within the cyclovirus Rep clade that may represent a novel group within the *Circoviridae*. Further sampling of CRESS DNA viruses found in arthropods and other invertebrates may help resolve phylogenetic relationships among members of the *Circoviridae*. Nevertheless, our phylogenetic analysis supports the idea that there are distinct groups of cycloviruses (*Dennis et al., 2018*).

It appears that the complete phylogenetic breadth of CRESS DNA viral diversity that has been reported to date circulates within arthropods and other invertebrates, which is analogous to what has been noted for RNA viruses (*Li et al., 2015*; *Shi et al., 2016a*, *2018a*). Few CRESS DNA Rep phylogenetic clusters are represented by viral sequences recovered from vertebrates, plants, or fungi alone (Fig. 1). This observation includes established CRESS DNA viral groups as well as novel sequences that have not been assigned to taxonomic groups. In addition, the vast majority of CRESS DNA sequences recovered from plants and fungi, including CEVs, fall near or within the closely related *Geminiviridae* and *Genomoviridae* clades. Vertebrate-associated CRESS DNA sequences that fall outside established groups have been mainly reported from fecal samples and most are intermixed with sequences that have been reported from invertebrates (Fig. 1). However, we identified one divergent clade that only included sequences from CRESS DNA viral isolates recovered from mammal feces and may represent a vertebrate-infecting lineage. More sampling, including blood or tissue samples as opposed to fecal samples, is needed to confirm this possibility. Despite the presence of plant-specific (*Geminiviridae* and *Nanoviridae*) and potentially vertebrate-specific viral lineages (genus *Circovirus*), most of the CRESS DNA viral diversity identified in vertebrates, plants and fungi is nested within the much broader genetic diversity of invertebrate-associated viruses.

## Related CRESS DNA viruses identified in disparate organisms

The phylogenetic analysis revealed many Rep sequences from disparate sources grouping together in the same clade (Fig. 1). Even when looking at broad source classifications, such as vertebrates, plants, arthropods and other invertebrates, few of the clades that fall outside of the established CRESS DNA viral groups represent isolates retrieved from similar sources. Moreover, the same "source intermixing" can be observed within established CRESS DNA groups (Figs. 2 and 3). CRESS DNA viral isolates representing members of the *Genomoviridae* are a notable example. Genomovirus genomes have been recovered from plants, fungi, vertebrates and arthropods; however, there is no clear separation of genomovirus groups based on the source (Fig. 2; Fig. S1). With such phylogenetic distribution, it is tempting to speculate about potential cross-species CRESS DNA virus transmission. However, since many CRESS DNA viruses have been identified through molecular assays alone, it is difficult to predict the host for most of these viruses, including those that fall within established CRESS DNA groups. Therefore, we cannot make inferences regarding horizontal CRESS DNA virus transmission. Furthermore, cross-species transmission between arthropod and vertebrate-infecting viruses has been deemed unlikely (*Dennis et al., 2018*). Nevertheless, available data suggest that closely related CRESS DNA viruses circulate among disparate organisms,
providing opportunities for cross-viral species interactions that may lead to recombination and the emergence of new viral species (*Krupovic, 2013*; *Roux et al., 2013*).

The detection of related CRESS DNA viruses in disparate eukaryotic organisms might be partly explained by vectored viruses and/or viruses infecting hosts that interact closely with other species in a similar niche. The two CRESS DNA viral families whose members infect plants and were not identified in our survey, namely *Geminiviridae* and *Nanoviridae*, are transmitted by hemipteran vectors where these viruses may be found in high titers (*Czosnek et al., 2017*; *Watanabe & Bressan, 2013*). In contrast to arthropod-borne animal-infecting RNA viruses and some vectored plant RNA viruses, it is thought that CRESS DNA plant viruses do not replicate or express genes in their vector (i.e., non-propagative transmission) (*Dietzgen, Mann & Johnson, 2016*). However, there is evidence showing genetic changes of a nanovirus within its aphid vector (*Sicard et al., 2015*). Moreover, some begomoviruses (family *Geminiviridae*) can replicate within their whitefly vector (*Czosnek et al., 2017*) and alter the whitefly feeding behavior to result in enhanced virus transmission (*Liu et al., 2013*). Therefore, there is evidence for complex cross-kingdom interactions between plant-infecting CRESS DNA viruses and their insect vectors.

In addition to recognized interactions between plant-infecting CRESS DNA viruses and their hemipteran vectors, CRESS DNA viruses may be present in systems where cross-kingdom species interactions are intertwined. We discovered CRESS DNA viruses in organisms that have been previously investigated for their role in model symbiosis systems, including fungus-insect and plant-insect systems. Interestingly, CRESS DNA viruses discovered in both fungus-farming insects investigated here, including bark beetle associated circular virus 1 (BbACV-1) and TACVs, were most closely related to members of the *Genomoviridae* (Fig. 2). Infection assays with *Sclerotinia sclerotiorum* hypovirulence-associated DNA virus 1 (SsHADV-1), the only genomovirus with a confirmed host (*Yu et al., 2010*), suggest that some genomoviruses are able to infect both fungi and insects. The primary SsHADV-1 host is a plant fungal pathogen; however, the virus is able to replicate in a mycophagous insect that it potentially uses as a transmission mechanism (*Liu et al., 2016*). Although we are not able to discern if BbACV-1 and TACVs infect either the farmer (insect) or the crop (fungus) in these ancient agricultural systems (*Mueller & Gerardo, 2002*), their discovery provides two additional examples where genomovirus-insect-fungi interactions are tightly connected.

Another notable example is the novel AaACV-1 cyclovirus, which was identified multiple times in all three ant species tested from the whistling thorn (*Vachellia* (*Acacia*) *drepanolobium)* ant-plant system in East Africa, with genomes recovered from each ant species sharing >98% PI. The three ant species tested (*Tetraponera penzigi*, *Crematogaster nigriceps*, and *Crematogaster mimosae*) live in an obligate mutualism with the acacia tree throughout its range (*Young, Stubblefield & Isbell, 1997*), protecting their host plants from mammalian herbivores, but also shaping aspects of the host environment such as plant-associated fungal communities, in a species-specific manner (*Baker et al., 2017*). Although the ants inhabit domatia (arthropod-occupied chambers) of the same acacia tree

species, each tree typically hosts a single colony at any point in time (*Palmer et al., 2000*). Opportunities for direct viral transmission between colonies of the same or different species are therefore likely to be restricted to infrequent antagonistic interactions between colonies occupying neighboring trees, so we did not expect to discover the same AaACV-1 virus in all three ant species. On the other hand, each tree typically hosts multiple colonies sequentially over its lifetime, providing the ants with shelter in domatia and food in the form of extrafloral nectar. The common host plant thus represents a good candidate mechanism for the circulation of AaACV-1 in these three different ant species. It remains to be determined what role, if any, AaACV-1 infection plays in this system. Interestingly, an endogenous cylcovirus Rep has been identified in another arboreal ant (*Pseudomyrmex gracilis*) (Fig. 3) (*Dennis et al., 2018*) suggesting that cycloviruses may be common in complex ant-plant symbiotic systems (*Clement et al., 2008*).

## Genomic fossil record supports widespread distribution of CRESS DNA viruses among invertebrates

Evidence from CEVs supports that CRESS DNA viruses infect or have infected a diversity of organisms, including hosts from four of the five supergroups of eukaryotes (*Belyi, Levine & Skalka, 2010*; *Dennis et al., in press*, *2018*; *Liu et al., 2011*). Here, we took advantage of previously reported CEVs and put them in a phylogenetic context with extant exogenous CRESS DNA viruses recovered from a wide array of organisms. Analyzed CEVs only included Rep sequences that were over 200 amino acids in length and did not contain any early stop codons or frameshifts. Therefore, analyzed CEVs potentially represent relatively recent CRESS DNA viral infections or functional elements co-opted by the host (*Dennis et al., 2018*; *Liu et al., 2011*). Notably, a CEV identified in the germline of the brown recluse spider *Loxosceles reclusa* (sequence ID: CVe. Loxosceles_reclusa.7, *Dennis et al., 2018*) is most closely related to the longjawed orbweaver circular virus 2 (LjOrbCV-2, accession MH545529) identified here, suggesting that viruses similar to LjOrbCV-2 infect spiders. Interestingly, all of the CEVs reported from vertebrates clustered within the genus *Circovirus*, whereas those reported from invertebrates were distributed across the CRESS DNA Rep phylogenetic tree (Fig. 1). Similarly, CEVs found in fungal genomes clustered near the *Genomoviridae* and *Geminiviridae* clades. Therefore, the available CEV data support that CRESS DNA viruses infecting vertebrates, plants, and fungi show a limited phylogenetic distribution compared to viruses found in invertebrates.

CRESS DNA-like endogenous viral sequences have revealed multiple insertions from related as well as distinct CRESS DNA viruses in various vertebrate host germlines (*Dennis et al., in press*). Multiple CEVs have also been reported from the same invertebrate species, with up to 19 sequences identified in a given host (*Dennis et al., 2018*) and our phylogenetic analysis supports that divergent CRESS DNA viruses can infect the same invertebrate host species (Fig. 1). For example, we were able to include in our analysis seven distinct Rep sequences previously identified as CEVs from brine flies (genus *Ephydra*) (*Dennis et al., 2018*), representing two species, *E. hidrans* ($n = 5$) and *E. gracilis* ($n = 2$). Surprisingly, none of the endogenized Rep sequences from brine
flies clustered close to each other. The closest phylogenetic neighbors to *Ephydra* Rep CEVs have been primarily recovered from aquatic invertebrates perhaps reflecting the ecology of these unique dipterans (Class Insecta) that live at the interface between aquatic and terrestrial habitats. Brine flies live in hypersaline alkaline lakes feeding on benthic algae and, whereas the larval stages are underwater, adults are considered terrestrial (*Herbst, 1980*). Note that one of the *Ephydra* CEVs clustered with a Rep representing a CRESS DNA virus recovered from another dipteran in its terrestrial phase (mosquitoes).

CRESS DNA-like endogenous viral sequences identified in parasitic invertebrates, including protozoans and nematodes, also demonstrate infection by distinct CRESS DNA elements. The presence of Rep-like sequences in the genomes of enteric protozoan parasites, including *Giardia* and *Entamoeba*, were noted more than 10 years ago (*Gibbs et al., 2006*). Further investigation of CEVs from these parasites showed that various Rep elements are transcribed, at least in *E. histolytica*, suggesting that the Rep has been co-opted for the benefit of the parasite (*Liu et al., 2011*). We were able to include seven Rep CEVs previously detected in *Giardia intestinalis* ($n = 2$) (*Gibbs et al., 2006*) and three species of *Entamoeba* ($n = 5$) (Fig. 1) (*Liu et al., 2011*). The two analyzed Rep sequences from *Giardia* clustered together, whereas sequences from *E. histolytica* clustered in two groups. Therefore, the unicellular parasite *E. histolytica* has been infected by at least two distinct CRESS DNA elements. Although we were not able to include multiple CEVs from the same nematode species, three out of four CEVs from parasitic nematodes clustered together, suggesting that there is a nematode-infecting lineage of CRESS DNA viruses or at least an ancestral one. Notably, extrachromosomal virus-like elements encoding a Rep have been reported from a free-living freshwater nematode (*Rebrikov et al., 2002*) and, thus, CRESS DNA-like elements may be more common in nematodes than previously recognized.

Circular Rep-encoding ssDNA viral infection of parasitic organisms transmitted through the fecal-oral route further exemplify the difficulties associated with predicting potential hosts for viruses identified in feces. There is a possibility that CRESS DNA viral sequences identified in vertebrate feces actually represent parasite-infecting CRESS DNA viruses since some parasites are ubiquitous and can be found in high numbers in fecal matter (*Oates et al., 2012*). For example, three of the CEVs from enteric protozoan parasites clustered close to CRESS DNA viral sequences retrieved from mammal feces (Fig. 1). Two of these were *Giardia* CEVs that clustered with FlyACV-6 and -7 identified here and rodent associated viruses (*Phan et al., 2011*). Since blow flies feed on fecal matter, FlyACV-6 and -7 may represent ingested viruses from feces that may infect either rodents or a parasitic protozoan host. Similarly, we detected a CEV in a rodent-infecting parasitic tapeworm, *H. microstoma*, whose Rep was most similar to cycloviruses recovered from rodents and cat feces (Fig. 3). With the available information, it is difficult to establish if the cat and rodent associated cycloviruses infect these mammals or co-occurring parasitic organisms. Alternatively, it is an intriguing possibility that parasitic eukaryotes can act as CRESS DNA viral vectors, which would result in tightly connected CRESS DNA virus-eukaryotic parasite-host interactions.

The detection of CEVs in a wide diversity of parasitic eukaryotes, including nematodes, protozoans, and arthropods indicates that the role of parasitic organisms in CRESS DNA viral ecology and evolution should be explored. Moreover, there are extant exogenous CRESS DNA viruses circulating in ectoparasitic arachnids (mites and ticks) (*Kraberger et al., 2018*; *Tokarz et al., 2018*; *Waits et al., 2018*; *Wang et al., 2018*). The detection of divergent RNA viruses in parasitic nematodes has also highlighted the need to further investigate the role of parasites in virus evolution (*Shi et al., 2016a*). The scarcity of data regarding parasite-associated microbes, including viruses, has been recognized and efforts are underway to try to address this knowledge gap (*Dheilly et al., 2017*). Exploration of undersampled invertebrate taxa and non-fecal samples from vertebrates will certainly provide more insights into the evolution of CRESS DNA viruses. In turn, the discovery of divergent CRESS DNA viruses will inform efforts investigating the genomic fossil record to better understand viral evolution and host biology (*Feschotte & Gilbert, 2012*; *Krupovic & Forterre, 2015*).

## CONCLUSION

Here, we described CRESS DNA viruses from terrestrial arthropods with a widespread phylogenetic distribution, including members of yet unclassified viral groups. The cosmopolitan distribution of CRESS DNA viruses, combined with the dynamic nature of these viruses, which seem to commonly exchange genetic information (*Kazlauskas, Varsani & Krupovic, 2018*; *Quaiser et al., 2016*), may help explain the unprecedented diversity recognized within recent years. It is important to note that our analysis underestimates the diversity within CRESS DNA viral genomes since we only included genome sequences associated with specific organisms, as opposed to environments (e.g., sewage, seawater), which would have added more than 200 Rep sequences to our phylogeny. In spite of this, it is clear that arthropods and other invertebrates harbor an extensive diversity of CRESS DNA viruses that dwarfs the genomic diversity observed in vertebrates, plants, and fungi. CRESS DNA viruses are emerging as a dominant and diverse group in the eukaryotic DNA viral world, with each report of novel genomes expanding the boundaries of this group. In contrast to other eukaryotic DNA viruses, CRESS DNA viruses are associated with a wide range of organisms across the tree of life, reflecting their ancient evolutionary history. Viral discovery efforts in undersampled taxa promise to reveal a more complete view of CRESS DNA virus diversity that will elucidate evolutionary linkages among these successful "genetic parasites" (*Koonin & Dolja, 2014*).

## ACKNOWLEDGEMENTS

We would like to thank students (Class of 2019) from the Laura Mercado Specialized School of Agro-ecology in San Germán, Puerto Rico for enthusiastically collecting specimens for this project during an outreach activity and Dr. José L. Agosto from the University of Puerto Rico, Río Piedras Campus for his assistance in conducting this activity. We also acknowledge Rachel C. Harbeitner, Dawn B. Goldsmith, Allison Cohen, Parker E. Jernigan, Sidney Fulford, and Carina Graham for their help with sample

processing. Thanks to Naomi E. Pierce (Harvard University) and Jon G. Sanders (University of California, San Diego) for facilitating access to ant samples from Kenya. Thanks to the CarBio team (http://www.islandbiogeography.org/participants.html) for their help collecting blow fly samples in the Caribbean.

### Funding

This work was funded through NSF Assembling the Tree of Life Program grant DEB-1239976 to Karyna Rosario and Mya Breitbart. Field work for blow fly collections in the Caribbean was funded through NSF grants DEB-1314749 and DEB-1050253 to Ingi Agnarsson and Greta Binford from the University of Vermont and Lewis & Clark College, respectively (Principal Investigators for Sohath Z Yusseff-Vanegas). Fungus-farming termites and ant samples from Africa were collected in the course of fieldwork funded by NSF grant DEB-1355122 to Corina Tarnita and Robert Pringle from Princeton University (Principal Investigators for Christopher CM Baker). The funders had no role in study design, data collection and analysis, decision to publish, or preparation of the manuscript.

### Grant Disclosures

The following grant information was disclosed by the authors:
NSF Assembling the Tree of Life Program grant: DEB-1239976.
Blow fly collection in the Caribbean was funded through NSF grants: DEB-1314749 and DEB-1050253.
University of Vermont and Lewis & Clark College.
Ant and termite collections in Africa were funded by NSF grant: DEB-1355122.

### Competing Interests

Mya Breitbart is an Academic Editor for PeerJ.

### Author Contributions

- Karyna Rosario conceived and designed the experiments, performed the experiments, analyzed the data, prepared figures and/or tables, authored or reviewed drafts of the paper, approved the final draft.
- Kaitlin A. Mettel performed the experiments, approved the final draft.
- Bayleigh E. Benner performed the experiments, approved the final draft.
- Ryan Johnson performed the experiments, approved the final draft.
- Catherine Scott contributed reagents/materials/analysis tools, approved the final draft.
- Sohath Z. Yusseff-Vanegas contributed reagents/materials/analysis tools, approved the final draft.
- Christopher C.M. Baker contributed reagents/materials/analysis tools, approved the final draft.
- Deby L. Cassill contributed reagents/materials/analysis tools, approved the final draft.
- Caroline Storer contributed reagents/materials/analysis tools, approved the final draft.

- Arvind Varsani analyzed the data, prepared figures and/or tables, approved the final draft.
- Mya Breitbart conceived and designed the experiments, analyzed the data, prepared figures and/or tables, approved the final draft.

## Data Availability

The genomes and replicons described here are accessible via GenBank accession numbers: MG917674 to MG917677 and MH545497 to MH545543.

## Supplemental Information

Supplemental information for this article can be found online at http://dx.doi.org/10.7717/peerj.5761#supplemental-information.

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
