# Peer review of "Virus discovery in all three major lineages of terrestrial arthropods highlights the diversity of single-stranded DNA viruses associated with invertebrates"

_PeerJ, doi:10.7717/peerj.5761_

## Round 0.1 · original submission · Minor Revisions

I am very pleased with your study since little is known about the diversity of ssDNA viruses in invertebrate hosts and you show that this is larger than the diversity found in other hosts. I agree with the three referees that the manuscript is well written, and the methodology and results are scientifically sound. I believe that the comments of the three reviewers are very appropriate. Especially those aimed at improving figures 1-3, also related to the criterion of demarcation of species that is somewhat confusing. Please attend to all comments and I will be happy to receive your response so that I can then accept the article. Thank you.

Reviewer 1 ·

Basic reporting

This paper is a thorough and well written analysis of 44 DNA CRESS viruses identified in different arthropods.

Experimental design

Complete genomes of 44 novel viruses were obtained by rolling circle amplification, restriction digest and cloning. The methods are appropriate and rigorous.
Phylogenetic analysis is also appropriate.

Validity of the findings

The data is robust.

Additional comments

It appears the sampling of arthropods was simply opportunistic. The authors could comment on what they believe would be an optimal sampling strategy to capture viral diversity.

Small circular DNA virus genomes have been identified as contaminants of various reagents (e.g. silica columns). What steps did the authors take to rule out the possibility that the detected viruses were actually from the putative host species and not perhaps an adventitious contaminant of a reagent?

·

Basic reporting

This manuscript describes the isolation of putative viral circular ssDNA sequences from a number of members of arthropod lineages, many of them not known previously to harbor ssDNA viruses. The novelty of these sequences is clearly shown, as is their diversity. As such I find that it is appropriate for Peer J. I am particularly impressed and gratified to be informed that CRESS DNA virus genomes have been found in arachnids, apparently for the first time.

I have a couple of suggestions for improvement. The main one is to emphasize the apparent chimerism in Rep protein sequences as described in Kazlauskas, Varsani and Krupovic, 2018. This chimerism is apparently highly prevalent in “unclassified” CRESS viruses which makes the construction of phylogenetic trees of whole Rep protein sequences problematic in terms of virus phylogeny. The authors might consider reducing the number of sequences presented, particularly in Figure 1 to reflect this. Figure 1 is overwhelming and it is not clear what information is to be conveyed in this figure. Having hundreds of separated lines for each sequence with nearly illegible text is excessive, in my opinion. I understand that the authors wish to present the diversity of their newly discovered sequences, but that could be done with considerably fewer sequences. I suggest simplifying this figure (and possibly Figures 2 and 3 as well), for the main text but providing the entire figure as a supplemental figure. I find the icons to be very useful and aesthetically pleasing, consider adding an icon label for the endogenous viruses in Figure 3.

It would be extremely useful to readers if a table with species demarcation percentages for different CRESS-DNA virus familes were presented. As currently in the manuscript it is very confusing to have different cutoffs for different families.

Experimental design

No comment in addition to those stated above.

Validity of the findings

No comment.

Additional comments

Specific comments:

I find the use of “PI” throughout a little confusing, consider defining “pairwise identity” in each section of the manuscript.

Lines 123-125. Consider expanding this statement to include chimerism.

Line 204. Consider either adding “with default parameters” or state whether alternative start codons or genetic codes were used to define ORFs.

Line 225: on (date) or in April 2018

Line 363 “of” species

Line 397 following: Please describe how is was determined that this 99% PI is not a contamination.

Line 437. Please describe how “non-significant” is defined.

Line 439. Consider reminding readers of the CEV abbreviation here.

Discussion. Please describe how virus families are defined.

539-540 – I find the “ancestral sentence” to be excessively speculative, particularly given recombination in Rep sequences.

Figure 2, Consider adding star to figure legend/on figure.

Footnote 3 of Table 1 Please clarify what parentheses mean.
Table 2. Please clarify why there are multiple accession numbers for some viruses.

Reviewer 3 ·

Basic reporting

The manuscript is clear, well written and logically structured. The introduction and background information are appropriate and demonstrate the context and relevance of the study. The structure is good and the figures are relevant, clear and nicely presented. The raw data can be accessed via genbank and the alignment can be opened and visualised with MEGA.

Experimental design

The manuscript and design of the study are scientifically and methodologically sound and fit well within the scope of the journal. The question of to what extent is CRESS DNA virus diversity represented in a sample of diverse invertebrate hosts is clear and the results are to provide useful information to expand knowledge about the diverse and still largely unexplored ssDNA virome of invertebrate hosts. The methods for sample processing and analysis are appropriate and clearly explained.

I just have a few minor comments to add where extra clarity/information may be added:

L157: What was the buffer for homogention? Further SM buffer after the three washes?
L187-192: How was it ascertained from the agarose gel whether the ends were sticky or blunt?
L196: what did you class as significant similarity?

L373: You previously described 78% similarity as a cutoff, but here you describe a novel virus FlyACV-2 as being 92% similar to one previously described. In a couple of cases including this one, and in the paragraph starting L388 it is not clear to me how exactly the boundary is decided between what you are classing as an isolate/specific virus species. A sentence could be added for clarity.

L402,408: could you provide examples of the shared genomic features.

Validity of the findings

The results of this exploratory study are sound. The sample size was covers a broad range of hosts/locations/time points, but the low level of replication is fine in this study as no assumptions were made about prevalence within hosts or the full extent of the diversity present in invertebrates. There are limitations to this study, as with any which explores viral diversity in a broad range of hosts ie, the sample size per host species is small, whether the viruses detected originate from true/active infections, possibility of contamination from infected food sources, etc, but these are discussed in the discussion, and although there is an amount of speculation in the discussion it is clearly presented as such.
The evolutionary history of the viruses discovered including using CEVs appears robust, yet is not overstated. The finding of distinct cycloviridae clades for example is informative and supported but the authors highlight the need for further sampling to resolve other relationships further.
The conclusions are appropriate. The authors have discovered a significant amount of previously unknown CRESS DNA virus diversity and the data indicates that the diversity in invertebrate hosts is far larger than that of CRESS DNA viruses infecting other taxonomic groups.

---

## Round 0.2 · accepted · Accept

Thank you very much for the revised version of your manuscript. I am satisfied with the response that you have given to reviewers. The text has significantly improved with the changes. I believe your manuscript is now ready for publication. Congratulations!

#